# Social Behavior and COVID-19: Analysis of the Social Factors behind Compliance with Interventions across the United States

**DOI:** 10.3390/ijerph192315716

**Published:** 2022-11-25

**Authors:** Morteza Maleki, Mohsen Bahrami, Monica Menendez, Jose Balsa-Barreiro

**Affiliations:** 1School of Industrial and Systems Engineering, Georgia Institute of Technology, Atlanta, GA 30332, USA; 2Institute for Data, Systems, and Society, Massachusetts Institute of Technology, 77 Massachusetts Avenue, Cambridge, MA 02139, USA; 3Division of Engineering, New York University Abu Dhabi, Saadiyat Island P.O. Box 129188, United Arab Emirates; 4MIT Media Lab, Massachusetts Institute of Technology, 75 Amherst St, Cambridge, MA 02139, USA

**Keywords:** behavioral analysis, COVID-19, governmental intervention, mask adoption, movement change, vaccine participation, non-pharmaceutical interventions, policy recommendations, social context, social behavior

## Abstract

Since its emergence, COVID-19 has caused a great impact in health and social terms. Governments and health authorities have attempted to minimize this impact by enforcing different mandates. Recent studies have addressed the relationship between various socioeconomic variables and compliance level to these interventions. However, little attention has been paid to what constitutes people’s response and whether people behave differently when faced with different interventions. Data collected from different sources show very significant regional differences across the United States. In this paper, we attempt to shed light on the fact that a response may be different depending on the health system capacity and each individuals’ social status. For that, we analyze the correlation between different societal (i.e., education, income levels, population density, etc.) and healthcare capacity-related variables (i.e., hospital occupancy rates, percentage of essential workers, etc.) in relation to people’s level of compliance with three main governmental mandates in the United States: mobility restrictions, mask adoption, and vaccine participation. Our aim was to isolate the most influential variables impacting behavior in response to these policies. We found that there was a significant relationship between individuals’ educational levels and political preferences with respect to compliance with each of these mandates.

## 1. Introduction

The emergence of the SARS-CoV-2 virus (COVID-19) has dramatically impacted the world over the last two and half years. Since the first cases were reported in the Chinese province of Wuhan in late 2019, the virus has been rapidly spreading across the globe. In March 2020, the situation was declared a pandemic by the World Health Organization (WHO) when the worldwide death count was 4291 and more than 118,000 cases were distributed across 114 countries. By mid November 2022, 633 million cases and 6.6 million deaths have been officially reported. The United States alone accounts for 97.5 million cases and 1.07 million deaths [1].

The COVID-19 pandemic caught most of the western countries devoid of answers. The still remaining large number of unanswered questions and the need to contain the impact of the pandemic forced most of them to adopt policies aiming to keep social distances and to restrict human mobility. Since the disease caused by the coronavirus is a type of respiratory disease, primary policies were based on two main *non-pharmaceutical interventions* (NPIs): mobility restrictions and mask mandates [2,3,4]. The forced restriction on human mobility aimed to minimize the number and frequency of interactions among people by suggesting to shelter at home except for essential reasons such as commuting to work, attendance to medical appointments, or basic shopping. Such policies were followed by closing non-essential businesses and limiting the capacity of essential ones in order to leave enough space for customers to keep social distance. Mask adoption mandates were enforced by most states since the emergence of COVID-19 virus and were adopted by almost all businesses and organizations. Even after the COVID-19 vaccine was majorly available after 2021, mask adoption was extremely encouraged by healthcare officials in the event of a rise in confirmed cases.

Many studies have evaluated the effectiveness of such interventions and their impact on urban agglomerations, where people live more concentrated. These studies have mostly used statistical modeling techniques to show the effectiveness of lockdown policies in mitigating the disease spread by reducing human mobility [3,4,5,6,7,8,9,10]. Although these studies indicate that adherence to social distancing is crucial in controlling the disease spread, Holtz et al. [11] show that ignoring the effects of social and geographical spillovers could negatively impact the effectiveness of such policies. This is followed by another line of research that pertains to modeling and predicting the disease spread across several scenarios [7,8,9,10,11,12]. These studies have made important contributions to improve and to implement more efficient responses.

Although all mentioned studies agree on the benefits of social distancing during the pandemic, they also indicate that the compliance with such policies shows significant differences among various socio-demographic groups, confirming the disproportionate impact of COVID-19 on some of them [7,13,14,15]. For instance, neighborhoods with lower income have been suffering from both higher infection rates as well as more negative impact on the employment rates. In addition to income, some social factors such as education levels and political belief explain differences in adherence to social distancing measures [16,17,18], although they are less important in comparison to the poverty level. For example, Painter and Qiu [17] showed that American residents in Republican counties were less likely to completely shelter at home after a state order. They also found that Democrats were less likely to respond to a state-level order when it was issued by a Republican governor relative to one issued by a Democratic one. Some other studies checked the effectiveness of face covering to prevent the spread of the coronavirus, showing significant differences among neighborhoods in adherence to mask adoption mandates [19,20,21].

Initially, we expected that NPIs were eventually adopted until until the implementation of a permanent solution in the form of vaccination took place. The eventual adoption of NPIs helped to minimize infection rates and to improve the response capacity of health system. The crucial objective was to massively reduce the number of deaths caused directly by the virus, but also to avoid the collapse of the healthcare system by keeping an optimal attention to all the people affected by the virus and/or any other ailments [1]. The successive approval of vaccines since the end of 2020 helped with minimizing the impact of the virus by reducing its severity.

Most of the western countries have experienced many difficulties in containing the virus. From the very beginning, various national strategies ranging from coexisting with the virus to its total suppression, the so-called *Zero COVID* strategy, initially adopted by Sweden and China, which are the most significant examples in both policies, respectively [22]. The unequal spatial patterns related to the spread and severity of COVID-19 were very noticeable over different geographies, but also considering the socio-demographic attributes of each community [23,24,25]. Significant differences were evident across continents, nations, regions, cities, and even neighborhoods. This has revealed the great territorial complexity associated with the virus and the emergence of vast territorial inequalities across multiple scales.

The actual impact of the virus goes far beyond the health issues, causing a great uncertainty about its effects in other sectors as well [26]. Thus, some scholars anticipate the impoverishment of large sections of communities, further emphasizing social inequalities. Beyond the great differences in terms of wealth (i.e., social coverage policies) and resources (i.e., health response capacity) with which different countries face this pandemic, the official number of infections reveals a precise indicator related to the impact of the virus across communities. The social dynamics behind the collective behavior help to understand understand the unequal distribution of the virus effects across the globe [27,28,29].

Regardless of the particular interventions and policies enforced by health authorities in each region, a crucial factor relates to the level of compliance with rules and recommendations by the people. Thus, adherence of citizens to the rules and their social behavior must be evaluated in order to better understand the real impact of policies. Collective responses should be investigated considering multiple factors and variables, which allows us to address the socio-spatial complexity behind the compliance with mandates. Among these factors, aspects related to individuals’ ideological and political preferences, level of income, educational levels, and socio-spatial determinants (rural vs. urban) must be considered as potential factors behind the virus spreading and people’s compliance with mandates.

From a spatial perspective, the virus impact was predominantly concentrated in cities during the first wave. According to the United Nations [30], urban regions became *ground zero* of the COVID-19 pandemic by allocating around 90 percent of the reported cases during the first weeks. In the United States, the impact of the virus in the central states, which are less populated, was delayed and more contained in the first months. There, compliance with the official rules was initially laxer because the virus was perceived as a distant threat.

In this paper, we analyze the association between a group of socioeconomic variables and the people’s response to three main governmental policies enforced by the American authorities for containing the pandemic: (a) mobility restrictions, (b) mask adoption, and (c) vaccine participation. Our aim is to isolate what were the most influential variables impacting people’s responses to these policies. The study results give us a better understanding of the collective behavior within human communities in the United States. Estimating the importance level of such factors is crucial for the design and implementation of more efficient policies in case of any other eventual emergency or threat in the future. This paper is structured as follows: Section 2 details material and methods, Section 3 presents the analysis and results, and Section 4 discusses the most significant findings.

## 2. Material and Methods

This section is divided into three sub-sections: Data Collection (Section 2.1), Data Parametrization (Section 2.2), and Data Processing (Section 2.3).

### 2.1. Data Collection

We collected information about healthcare and socioeconomic indicators from various data sources. Healthcare indicators refer mainly to the health system’s capacity [31], whereas socioeconomic indicators refer to aspects related to educational level [32,33], political preference [34], and income level [32] within communities. We also included the data related to how people are spatially distributed in each geographical unit [35], which can help to better understand the social dynamics across different scales [36,37]. In addition, a number of indicators related to the COVID-19 impact were particularly considered within a third group containing information about epidemiological data in terms of incidence and tests, in addition to information related to (a) mobility restrictions, (b) mask adoption, and (c) vaccine participation [31,38,39].

Altogether, we considered 25 indicators and organized them into six groups. For a simpler analysis, each group was assigned a unique code, structured as an XX–YY or XXX–YY code, where XX or XXX refer to each of the six major groups, while YY identifies each individual indicator. The particular codes of each major group stand as follows: COVID-19 (C19), Education Level (EL), Healthcare Capacity (HC), Economic Level (EC), Political Preference (PP), and Population Settings (PS). A complete list of these groups and indicators, and the relevant metadata (i.e., descriptor, sample number, time frame, and value range) are shown in Table 1 and Table 2.

The value of the respective indicators is quantified by counties across the United States. Administratively, the country has 3143 counties (and county equivalents) showing very significant differences in both size and population among them. The largest counties are located in the western sector, while the most densely populated counties are located on both coastal shorelines.

Datasets are collected and extracted from different sources, including both official repositories (i.e., the US Census Bureau [35]) and unofficial ones, ranging from social networks (Facebook’s Data for Good Initiative [38]) to media surveys (New York Times [39]). The time frame of this study corresponds to a 2-week period, between 2 July and 14 July 2020. For the official datasets, we opt for the most recent date that corresponds to the end of 2019 or 2020. In order to provide accessibility to all the data used in this study, we implement a publicly available GitHub repository where we store all datasets used in this study (see the Data Availability Statement). The final dataset contains information on 25 individual indicators (columns) for 3143 counties (rows).

A brief description of the individual indicators and the way this is organized in six variable groups is shown below:

#### 2.1.1. COVID-19 Data (C19)

The C19 group contains information related to the COVID-19 impact, but also the people’s responses to that. We define four subgroups of data within: (a) epidemiological data, (b) mask adoption within a certain distance, (c) mobility restrictions, and (d) data related to vaccine participation.

Regarding (a), we include information from four different individual indicators related to incidence and testing factors. Incidence is quantified based on the number of new COVID-19 related cases (C19-CC) and deaths (C19-DD). In both cases, data corresponds to the total number of cases/deaths officially reported for a 7-day period. Data are sourced from US COVID-19 Atlas [31], where these are updated on a daily basis. In addition, the number of tests performed (C19-TT) and the testing capacity (C19-TC) are indicators pertaining to testing factors and healthcare capacity.

Regarding (b), we include six different indicators related to the mask adoption, which is defined as the frequency that people were using the mask at that time. These data were extracted from a survey conducted by the New York Times between 2 July and 14 July 2020 [39]. We analyze 250,000 responses from US citizens that explicitly replied to this question: “How often do you wear a mask in public when you expect to be within six feet of another person?”. They could take five different options in an increasing level of their adherence to mask adoptions, ranging from never (C19-MN) to always (C19-MA). The survey also includes the age and gender of the surveyed people. We implement a mask adoption total score (C19-MC) that combines the response to each of the five potential replies (see Figure 1).

Regarding (c), we define two indicators. The first one (see Figure 2) estimates mobility changes with respect to a baseline before the pandemic emergence, in February 2020, when no mobility restriction orders had been issued (C19-MO). The second indicator refers to the percentage of people staying at home with respect to the same baseline (C19-PT). Both indicators showed high spatial heterogeneity due to the policies adopted by the different states/counties. Data related to these indicators are sourced from Facebook’s Data for Good Initiative [38]. This dataset provides information about human mobility in comparison to a baseline period that predates most social distancing policies. Based on this, we estimate how people replied to mandatory recommendations related to social distancing and mobility restrictions during the period that ranges from 2 to 14 June 2020.

Regarding (d), we use one single indicator, i.e., COVID-19 vaccine (C19-VC), which has been derived from the vaccine participation dataset for 1 September 2021 in this study (see Figure 3). Data are sourced from the COVID-19 vaccine participation tracking website of Georgetown University, which assembles data from the Center for Disease Control and Prevention (CDC) and the official reports provided by the different US States [31]. For data processing, vaccine participation data has been normalized to account for differences in population within each age group.

The reason we considered 1 September 2021 vaccine participation data is to take into account the fact that at the early stages of vaccine participation (mid December 2020 to March/April 2021) vaccines were not available at all locations and counties, therefore measuring the impact of socioeconomic variables on vaccine participation would clearly be biased. On the other hand, considering vaccine participation data at the later stages (November 2021 to mid 2022) would also bias the results in that in these time-frames vaccines have already been available for people to take for a long time (about 6 months or more) and taking vaccines is not out of choice, but rather because of employer mandate, travel requirements, etc., which clearly does not represent the true intention of people in regards to vaccine participation.

#### 2.1.2. Educational Level (EL)

We differentiate two simple groups according to their educational background (see Figure 4: the percentage of adults with less than a college degree (EL-LC), and those holding a college degree or higher (EL-MC). These datasets are sourced from the US Census. In addition, we evaluate the amount of federal investment in education at the county level. The data was obtained from the US Government Data Lab [33].

#### 2.1.3. Health System Capacity (HC)

We use three indicators related to the capacity of the health system. These refer to the total number of available beds in hospitals (HC-HB), their occupancy rates (HC-HO), and the percentage of essential workers in healthcare (HC-PW). This data has been obtained from the COVID Atlas [31].

#### 2.1.4. Economic Level (EC)

We define three indicators related to poverty and unemployment. Poverty estimates (EC-PO) refers to the percentage of people with income levels lower than 14,097 USD per year [40]. The unemployment rate (EC-UN) refers to the percentage of people that were not employed. Median of Income (EC-IN) is measured by estimating the median of income for each US county. These data are sourced from the US Census Bureau [35].

#### 2.1.5. Political Preference (PP)

We define the political preference based on the results of the last US presidential elections held in 2020. We use a binary variable to specify which party won the election per US county. Democratic Party Win (PP-DW) is a binary variable indicating whether the democratic party had a higher vote share for a particular US county (see Figure 5). Data are sourced from the MIT Election Lab [34].

#### 2.1.6. Population Settings (PS)

We evaluate two indicators within population settings: population density (PS-PD) and percentage of rurality (PS-RL) [41]. The first indicator (PS-PD) refers to the average population by area unit. This is expressed by the number of people per square mile. The percentage of rurality (PS-RL) refers to the relative number of people living in rural regions. Data is sourced from the US Census Bureau [35].

### 2.2. Data Parameterization

In Table 2, we show the range of values, from a lower to an upper limit for the time frame considered. The values for each indicator are associated with counties. For data analysis, all the indicators, including dependent variables are between 0 and 1. They were either originally between in this interval (e.g., EL-MC and C19-VC) or, they have been transformed to this same interval using a min-max normalization. The only exception is the Mobility Restrictions’ variable (C19-MO), which has negative values and ranges already between −1 and 1. This is used without any changes. The reason for this normalization is to create a simple coefficient that eliminates bias from variables with too large or too small values. It is important to note that this normalization is only applied for regression analyses and not for spatial mapping visualizations or correlation analysis in the Appendix A.

### 2.3. Data Processing

The indicators considered allow us to assess people’s responses and the impact of the pandemic, in both directions. We apply multiple regressions to check the weight of the individual indicators. We define a series of explanatory variables for each policy adopted by authorities, i.e., (a) mobility restrictions, (b) mask adoption, and (c) vaccine participation. The analysis is conducted using various libraries for statistical analyses in R (e.g., corrplot, ivreg, data.table, and tidyverse). Data visualization is carried out in *Tableau*, and regression analysis tables are produced using the *R package* named *Stargazer* [42].

To avoid multicollinearity effects, we apply a correlation analysis on the complete list of variables shown in Table 1 and Table 2. The results are shown in Figure A1 in the Appendix A correlation value of 0.8 or greater was used to identify highly correlated variables, which were subsequently eliminated to avoid multicollinearity. At the end, we obtain a list of the following 12 variables: C19-CC, C19-MC, C19-MO, C19-VC, EL-MC, EL-FI, HC-HO, HC-PW, EC-PO, EC-UN, PP-DW, and PS-PD. Therefore, the resulting dataset contains data for 3143 counties (rows) and 12 indicators (columns). We conduct our study over this dataset. It is important to consider the fact that some confounding variables can exist in the pool of our independent variables. In particular, the political preference variable that has been used as one of important independent variables in our study, could be potentially associated with socio-economic variables. To minimize any bias in our results, we checked for this. As shown in Figure A1–Figure A4 in Appendix A, none of the dependent variables in our study are highly correlated with political preference, therefore eliminating bias and making political preference a suitable candidate for being used as an independent variable in this study. We also note that this variable has been recently used as an independent variable in other recent studies as well [17,18].

## 3. Analysis and Results

In this section, we carry out a multiple regression analysis to discover significant variables associated with people’s response to each of the three individual interventions, namely: mask adoptions, mobility restrictions, and vaccine participation. While the causal processes that produce the results are complex, in order to get more meaningful results we then add an instrumental variable to the regression models. The analyses results provided in this section are organized as follows: Multiple Linear Regression (Section 3.1), Instrumental Variable Regression (Section 3.2), and comparison between both methods (Section 3.3).

In the following subsections, we introduce all the variables considered in both regression models (multiple and instrumental). After that, we show which ones are the more significant ones for the respective models. We briefly discuss about if these results present some logic by providing some arguments.

### 3.1. Multiple Linear Regression

Multiple linear regression (MLR) is a well-known and broadly applied ordinary least squares (OLS) based statistical technique that uses several explanatory variables to predict the outcome of a response variable. The goal of multiple linear regression is to model the linear relationship between the explanatory (independent) variables and a response (dependent) variable. This model has been utilized by many recent research works aiming to understand the dynamics of social behavior and collective response to public health policies in the context of the COVID-19 pandemic [11,17,19].

The regression models we implemented are based on the following equations (Equations (1)–(3)):(1)$$Mobility\;Restrictions=\alpha_{MO}+\beta_{MO}\times PP-DW+\gamma_{MO}\times EL-MC+\delta_{MO}\times CONTROLS+\epsilon_{MO}$$
(2)$$Mask\;Adoption=\alpha_{MC}+\beta_{MC}\times PP-DW+\gamma_{MC}\times EL-MC+\delta_{MC}\times CONTROLS+\epsilon_{MC}$$
(3)$$Vaccine\;Participation=\alpha_{VC}+\beta_{VC}\times PP-DW+\gamma_{VC}\times EL-MC+\delta_{VC}\times CONTROLS+\epsilon_{VC}$$

In these equations, CONTROLS refer to the controlling variables in each regression model. These variables are used to account for hidden effects and confounding variables so the final analysis of results has a low bias level. The following variables are used as CONTROLS: C19-CC, HC-PW, HC-HO, EC-PO, EC-UN, and PS-PD. Since, these are not the target variables of this study, they are included in the models to control for their effects. The parameter refers to the intercept value in each regression equation, refers to the coefficient of the PP-DW variable, refers to the coefficient of the EC-MC variable, and finally refers to the aggregate of controlling variables’ coefficients in each regression. It is important to note that all the variables used represent a collective response at a county level.

The results of all three multiple regressions are displayed in Table 3, Table 4 and Table 5. There, rows show the different explanatory variables with regard to the dependent variable discussed in that specific table. In order to understand the effect and significance of each independent variable of interest on the outcome, we use the step-wise variable addition approach. We create three different models for each dependent variable and we add controls and variables of interest incrementally to evaluate their contribution to the analyses. In the regression tables, each column refers to a model used for analysis.

For each regression analysis, we included independent variables: PP-DW, PS-PD, and EC-PO in Model 1. In Model 2, we added EL-MC to the list of predictors. In Model 3, all the independent variables and controls were included. The reason for this step-wise addition of variables is that we implement a forward selection algorithm in model selection. We first start with a null model and then add the independent variables to the model one-by-one. We estimate the R^2^ for each resulting regression. If after addition of a variable, R^2^ of the regression was not improved, then we eliminate that variable from the regression. For each regression model, only the most significant variables that provide the best R^2^ were selected, resulting in Models 1, 2, and 3. There are exceptions to this general rule as in Table 3, where the incorporation of EL-MC did not result in a significant improvement in the R^2^ of Model 2. However, since EL-MC is the main variable of analyses here, we refrained from eliminating it being included in Models 2 and 3.

For discussion purposes, coefficients in Model 3 of each regression table are considered. The coefficients in Models 1 and 2 are provided just to show the process through which the step-wise addition of variables took place and final models’ variables were selected.

#### 3.1.1. Mobility Restrictions (C19-MO)

As shown in Table 3, political preference (PP-DW) is statistically significant in all three models after controlling for several socioeconomic factors and COVID-19 related variables. This means the counties with more democratic leaning political preference show less movement under all three regression models. This is in line with previous studies by confirming how those regions with higher political preference towards the Democratic party showed more adherence to the social distancing and shelter at home orders [17].

Education level (EL-MC) is introduced in Models 2 and 3. While it is not statistically significant in Model 2, after adding control variables in Model 3 it becomes statistically significant which shows that the counties with higher levels of education, would have been expected to observe more movement during the studied time frame with respect to the baseline before the pandemic started. Moreover, from the resulting coefficients for population density (PS-PD), it is evident that the people living in more densely populated regions reduced substantially their mobility much more than the rest of the people. In addition to negative association of the confirmed COVID-19 cases (C19-CC), this positive adherence could arguably be a result of differences in their higher perceived risk of exposure and infection.

In short, regarding the significance of each variable in Model 3, results show that all variables considered are statistically significant, with PS-PD, PP-DW, C19-CC, EC-PO, and HC-HO having a negative association with the change in movement of individuals (C19-MO), and EL-MC, HC-PW, and EC-UN having a positive association.

#### 3.1.2. Mask Adoption (C19-MC)

The results in Table 4, show that political preference (PP-DW) and education level (EL-MC) are statistically significant showing a positive association with mask adoption (C19-MA), indicating that in the counties that voted for the Democratic party in 2020 and/or have higher levels of education, residents wear masks more frequently.

Based on the resulting coefficients for population density (PS-PD), similar to the change in movement (C19-MO), it is evident that the people living in areas with higher population density, used their masks more to protect themselves as they had a higher chance of encounters with other infected residents. This is also true about the variable representing the COVID-19 confirmed cases (C19-CC). As the number of infected people in a county increase, rate of mask usage increases, which can be associated with their perception of higher risk of infection.

#### 3.1.3. Vaccine Participation (C19-VC)

The results shown in Table 5 indicate that political preference (PP-DW) and education level (EL-MC) are statistically significant showing a positive association with vaccine participation rate (C19-VC). Similar to the mask adoption rate (C19-MC), the counties that voted for the Democratic party in the 2020 Presidential election, would have been expected to observe higher vaccine participation rates (C19-VC). This is also true for counties with higher level of education (EL-MC), which were more responsive to participate in the vaccination during the first months.

### 3.2. Instrumental Variable Analysis

Analysis based on instrumental variables (IV) is a method for uncovering causality in socioeconomic research. This is a powerful tool for finding out whether there exists a causal relationship between two variables by considering an instrument. In the previous subsection, the outcome variables C19-MC, C19-MO, and C19-VC were analyzed using multiple linear regression.

In this subsection, we investigate the role of higher education on each of the dependent variables. Specifically, we aim to investigate if there is a causal relationship between the level of education and complying with governmental mandates during the pandemic. With this in mind, we used the Federal Investment in Education (EL-FI) as an instrument in our analysis to form Instrumental Variable regression. EL-FI satisfies the relevance and validity assumptions in our study (see Appendix B). It satisfies the relevance assumption because it is highly associated with the endogenous variable EL-MC. It satisfies the validity assumption because it passes the Wu–Hausman for weak instruments, which shows that EL-FI and hidden variables impacting the outcome of interest (compliance with mandates) are not correlated. Additional explanations with more details regarding the causal diagram of the IV regression, equations, and assumptions are provided in Appendix B.

We use the three following regression models:(4)$$Mobility\;Restrictions=\alpha_{MO}+\beta_{MO}\times PP-DW+\gamma_{MO}\times EL-MC+\delta_{MO}\times CONTROLS+\epsilon_{MO},\;Z=EL-FI$$
(5)$$Mask\;Adoption=\alpha_{MC}+\beta_{MC}\times PP-DW+\gamma_{MC}\times EL-MC+\delta_{MC}\times CONTROLS+\epsilon_{MC},\;Z=EL-FI$$
(6)$$Vaccine\;Participation=\alpha_{VC}+\beta_{VC}\times PP-DW+\gamma_{VC}\times EL-MC+\delta_{VC}\times CONTROLS+\epsilon_{VC},\;Z=EL-FI$$
where Z is the instrument for each of the IV regression models.

The main reason for conducting IV regression analysis is to isolate the causal impact of education level (EL-MC) on each of the three mandates. We conduct a step-wise IV regression for Equations (4)–(6). The results of all three multiple IV regressions are displayed in Table 6, Table 7 and Table 8. In each IV regression table, Model 1 is the simplest, capturing merely the causal impact of EL-MC without any control variables considering EL-FI as an instrument. Model 2 adds the PP-DW variable to the existing IV regression of EL-MC on target mandates to better understand its impact on the causal relationship of interest. Model 3 includes all non-explicit control variables in addition to EL-MC, PP-DW, still considering EL-FI as the instrument.

#### 3.2.1. Mobility Restrictions (C19-MO)

As Table 6 indicates, PP-DW has a negative impact on C19-MO in Model 3, and it is statistically significant. Since Model 3 is the only model wherein we introduced all controls in addition to PP-DW and EL-MC variables in the IV regression, the results can be accepted with more confidence compared to Models 1 and 2. The results are similar to those of OLS regression estimation. More details about this are discussed in Section 3.3.

EL-MC is introduced in all of the models and it is initially statistically significant in Models 1 and 2. Once PP-DW is introduced along with all control variables in Model 3, the estimated value loses its significance, discarding the notion that EL-MC has a causal impact on the outcome variable C19-MO using EL-FI as the instrumental variable.

In summary, the results suggest that political preference (PP-DW) had a causal impact on mobility and the resulting movement patterns (C19-MO) as well as compliance with social distancing orders. On the other hand, education level (EL-MC) did not have a causal impact on mobility (C19-MO).

#### 3.2.2. Mask Adoption (C19-MC)

Table 7 shows the IV regression’s results for the variable C19-MC. The results show that PP-DW has a positive impact on C19-MC in Model 3, being statistically significant. Since PP-DW is introduced in Models 2 and 3, and all the control variables are present in Model 3, we can accept such statistically significant and positive results for its coefficient in Model 3. According to this, the counties that voted for the Democratic party are expected to have higher rates of mask adoption during the time period here considered. These results are similar to the results of OLS regression. More details about this are discussed in Section 3.3.

EL-MC is introduced in all models. It is initially statistically significant in Models 1 and 2, but after the introduction of PP-DW and all control variables in Model 3, the estimated value loses its significance, discarding the notion that education level (EL-MC) has a significant causal impact on the mask adoption of individuals (C19-MC).

#### 3.2.3. Vaccine Participation (C19-VC)

Table 8 shows the results of IV regression for the vaccine participation as the target variable (C19-VC). In Models 2 and 3, we can observe how PP-DW has no impact on C19-VC. This finding is different from the results produced by OLS regression in Table 5. More details about this are discussed in Section 3.3.

EL-MC is introduced in all of the models, being statistically significant (although not at the same level) in Models 1 and 3. After PP-DW is introduced in Model 2, the estimated value for the EL-MC coefficient loses its significance. However, after the introduction of all control variables along with PP-DW and EL-MC in Model 3, a positive and statistically significant coefficient for EL-MC is estimated. These results indicate that the education level (EL-MC) had a causal impact on vaccine participation (C19-VC), such as was mentioned before.

### 3.3. Comparison between OLS and IV Regression Results

In this section we conduct a comparison study of the results obtained by both OLS and IV methods for the three target variables. Table 9 shows a summarized view of the results of analyses only for Model 3 as it includes the most complete set of the control and independent variables.

The Wu–Hausman’s test [43,44] evaluates the consistency of an estimator when compared to an alternative—less efficient estimator—which is already known to be consistent. It helps one evaluate if a statistical model corresponds to the data. In case of rejection, as is the case in all IV regressions in this study, the results obtained in the OLS regressions with the same dependent and independent variables are more reliable and should be accepted. In the following, more explanations regarding comparisons between OLS and IV regression for each target variable are provided.

#### 3.3.1. Mobility Restrictions (C19-MO)

Table 9 shows that the PP-DW’s coefficients in both OLS and IV regressions are statistically significant. The slightly negative values confirms an inverse—but minimal in magnitude—association between PP-DW and C19-MO. Since the Wu–Hausman’s test is rejected for the IV regression, the OLS results will be accepted.

The EL-MC’s coefficient is statistically significant in the OLS regression in contrast to the one obtained in the IV regression for Model 3. However, since the Wu–Hausman’s test is rejected, it can be inferred that the OLS results can be accepted. Although there seems to be no causal relationship between education level (EL-MC) and change in movements (C19-MO), we can argue there is a strong association between these both variables.

#### 3.3.2. Mask Adoption (C19-MC)

According to Table 9, the PP-DW’s coefficients in both OLS and IV regression are statistically significant and very close to each other in terms of magnitude. However, since the Wu–Hausman’s test is rejected, the OLS result will be accepted and considered as final when determining the association between mask usage (C19-MC) and political preference (PP-DW) at a county level.

The EL-MC’s coefficient is not statistically significant in the IV regression, but it is in the OLS regression methods after the control variables are introduced. Since Wu–Hausman’s test in IV regression is rejected, OLS result will be accepted, which means that although education level (EL-MC) does not have a causal impact on mask usage (C19-MC), there is a strong correlation between the two variables.

#### 3.3.3. Vaccine Participation (C19-VC)

Based on the results shown in Table 9, the PP-DW’s coefficient is statistically significant in the OLS regression, in contrast to the IV regression. However, the association between PP-DW and C19-VC in the OLS regression is relatively small in terms of its magnitude. On the other hand, the EL-MC’s coefficient in both OLS and IV regression against C19-VC shows a strong relationship between both variables and some causality effects, indicating that the education level (EL-MC) is positively correlated with vaccination participation (C19-VC).

In summary, in all regression models, we included education level (EL-MC) and political preference (PP-DW) as the main independent variables to identify their association with behavioral response variables. In order to improve the explanatory power of the models, we additionally included relevant socioeconomic and healthcare infrastructure related variables before and during the COVID-19 pandemic. Although not all independent variables are statistically significant, the presence of such controlling variables significantly helped to check the true impact of the main independent variables and to minimize the bias effect. In addition, we used an IV regression approach to better understand the relationship among variables and to raise our confidence in capturing the causal impact of education on compliance with public health related mandates. Therefore, although some regression models might not possess significantly high $R^{2}$ values, integrating their results with the IV regression approach enriches the insights presented in this study.

## 4. Discussion and Conclusions

We divide this section into three minor sub-sections as follows: (Section 4.1) Summary, (Section 4.2) Limitations, and (Section 4.3) Implications and future directions.

### 4.1. Summary

In order to better understand the COVID-19 pandemic, one must analyze its incidence and impact using statistical tools and analytical approaches. Noticeably, the large variability in the virus impact across regions requires the consideration of a vast number of variables related to the physical mechanisms behind the virus spread and the complex behavior of human societies.

In this study, we attempt to deal with this social complexity by conducting a multivariate research of the uneven spatial impact of COVID-19 across the United States. For that, we analyze the correlation between the COVID-19 incidence and a diverse group of variables related to the healthcare system in addition to other socioeconomical variables such as people’s educational level and political preferences. These variables are jointly analyzed with people’s responses to the three major interventions adopted by the U.S. health authorities for containing the pandemic: (a) mobility restrictions, (b) mask adoption, and (c) vaccine participation.

According to our results (summarized in Table 9), investigating socioeconomical factors is crucial for understanding spatial differences in the COVID-19 pandemic. Factors related to people’s political preferences (PP-DW) were one of the most influential variables for understanding the responses to mask adoption (C19-MC) and mobility restrictions (C19-MO), whereas educational level (EL-MC) was more important for understanding the uneven engagement in the vaccine participation (C19-VC). Other variables such as the relative severity of the COVID-19 impact, which can be estimated using proxies such as the level of hospital occupancy by COVID-19 patients (HC-HO) in relation to the percentage of essential workers in hospitals (HC-PW) were also significant, but much smaller in magnitude than the education level (EL-MC).

Political preference (PP-DW) demonstrates significant differences in people’s responses to COVID-19 mandates across the US counties. Republican counties tend to register higher mobility and lower intention for mask adoption. However, unlike what has been reported before [45], political preference alone did not explain variations in vaccine participation rates in our models. This might be explained by the fact that we accounted for a wider range of control variables and emphasized on education levels using federal investment on education (EL-FI) as instruments in our analyses. The results we obtained demonstrate that although political preferences changed the perception of the pandemic, it only did so to a limited extent.

Some population setting attributes such as population density (PS-PD) were also relevant. Table 9 shows how population density and mask adoption are significantly associated (0.835). Obviously, the particular COVID-19 transmission mechanism could have led to a heightened perception of the disease in urban regions, at least during the first months when the vast majority of the infections were located in dense urban areas.

People’s educational level was found to be a decisive factor for understanding the spatial variation in the engagement in the vaccination against the virus. We found a strong relationship (0.654) between those counties with a larger predominance of highly educated residents (EL-MC) and the rate of vaccinated people (C19-VC) according to the OLS regressions shown in Table 9. This was further confirmed by an IV analysis where an increase of one unit in educational level caused an increase of 1.774 units in vaccine participation.

In this multi-factorial analysis, we also investigate and analyze the correlation between variables. For instance, according to results presented as a pairwise correlation table in Figure Figure A1, the Republican counties present higher rurality rates and, therefore, lower population densities. Also in these counties, the reported average household incomes and education levels are lower. These counties tend to present lower COVID-19 vaccine participation rates [46,47] and higher rates of COVID-9 incidence, at least during the time period here considered. On the other hand, we find that the Democratic counties experienced on average less harm from COVID-19 which is confirmed by other research as well [41].

### 4.2. Limitations

This study presents some limitations. The most important refers to the number of indicators considered. We attempted to reduce the social complexity related to the social behavior within communities to a short number of indicators. This is determined by the data availability during the time window considered. We collect relevant data related to collective behavior in relation to vaccine participation, mask usage, and mobility patterns in the first months of the pandemic. Although this approach could lead to an oversimplification of the human behavior, our results shed light onto the most basic relationships between different socio-economic variables and the response to the main interventions adopted by US authorities. In this way, we conduct a optimal trade-off between the number of variables and the significance of the models implemented.

We must note that our data were retrieved from different data sources. These data were collected using different methodologies. Most of these data were constrained to a very short time window at the beginning of the pandemic. Thus, people’s responses to the COVID-19 pandemic must be contextualized within the early months. This limitation is common among early studies trying to model people’s behavior during the first months of the pandemic [11,12,21].

It is also important to note that performing behavioral analyses and modeling collective behavior requires more detailed microscopic data. Access to such detailed datasets at the individual level would potentially enhance the accuracy and interpretation of our results. That being said, macroscopic datasets like the ones used in this study are still extremely useful in identifying general patterns (e.g., in mobility [48,49]). Moreover, although many governments and institutions offered open datasets during the pandemic, many data presented important limitations and restrictions related to spatio-temporal resolution, privacy concerns, etc. Our analysis was conducted at a county level by merging information collected from different data sources. Our results are in accordance with other research studies, but we also observed some divergences with other ones conducted at different spatial scales [37].

### 4.3. Implications and Future Directions

In this study, we investigated the causal relationship between social context and the level of adherence to the COVID-19 interventions in the United States. The results are of high relevance for better understanding social behaviors and to implement more efficient policies in emergency situations such as the current pandemic and future ones. Our findings help to explain the heterogeneous impact of COVID-19 on different communities, and thus to adopt policy implications for encouraging the public to comply with the public health policies. Our results are especially significant regarding the vaccination programs and may help public health authorities to adopt more efficient policies in order to increase the vaccination rates through educational programs.

This study progresses in line of some recent requests [50]. Our results can be extrapolated to other regions. Similar studies were already implemented in other countries [2,3,4,9,24]. For instance, Flaxman et al. [9] show that non-pharmaceutical interventions such as social distancing encouragement, banning public events, ordering school closures, and ordering lockdowns had a significant impact on preventing or slowing the spread of COVID-19 in 12 European countries including Austria, Belgium, Switzerland, Spain, Italy, etc., in early 2020. The number of deaths as a result of COVID-19 was significantly reduced in countries with such policies compared to other countries with no such mandates in place. Carballosa et al. [24] simulated the impact of mobility restrictions among municipalities in Spain. They observed that the confinement of the economically non-active individuals (elderly and children) may result in a significant reduction of risk, showing quite similar effects to the lockdown of the total population.

This research can be expanded by increasing the number of variables in the model. Another possible extension is to apply spatial econometric models to the current data in order to better understand underlying relationships among the variables used in this study.

## 5. Disclaimer

The period for when C19-MC and C19-MO were collected ranges between 2 July to 14 July 2020. Ideally, comparison in mobility patterns could have been made with a similar period a year before (to discard seasonal variations). However, data for June 2019 was not available to the authors.The results in this paper were briefly presented at the poster session as M. Maleki, M. Bahrami, M. Menendez and J. Balsa-Barreiro (2022). Investigating the causal impact of education levels on compliance with mandates, in 8th International Conference on Computational Social Science (IC2S2), Chicago IL, USA, 19–22 July.

## Figures and Tables

**Figure 1 ijerph-19-15716-f001:**
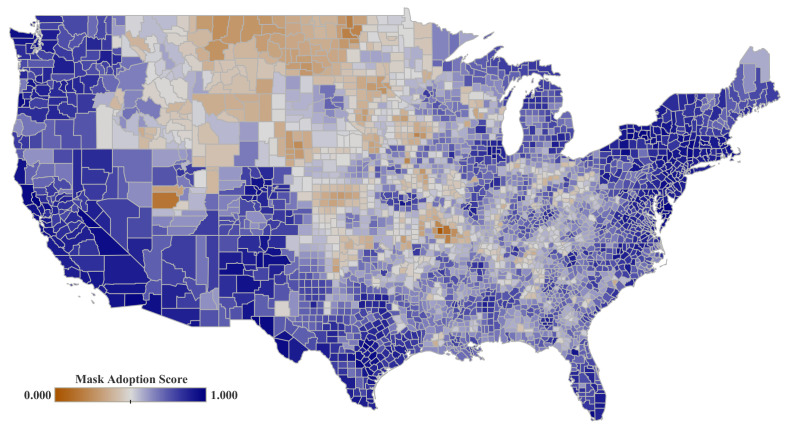
Mask adoption for the period from 2 to 14 June 2020. Data score is estimated from a New York Times survey for 250,000 people and extrapolated to the whole United States [39]. Data is spatially aggregated by counties for the United States mainland. Data scores range from never (0) to always (1).

**Figure 2 ijerph-19-15716-f002:**
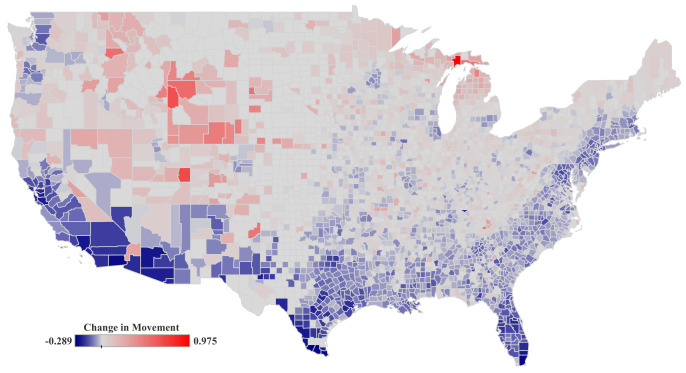
Mobility changes for the period from 2 to 14 June 2020. We measure the relative mobility restrictions (C19-MO) compared to baseline. Data is spatially aggregated by counties for the United States mainland. Data scores range from reduced (−0.289) to increased (0.97) mobility. Value 0 stands for the same mobility level.

**Figure 3 ijerph-19-15716-f003:**
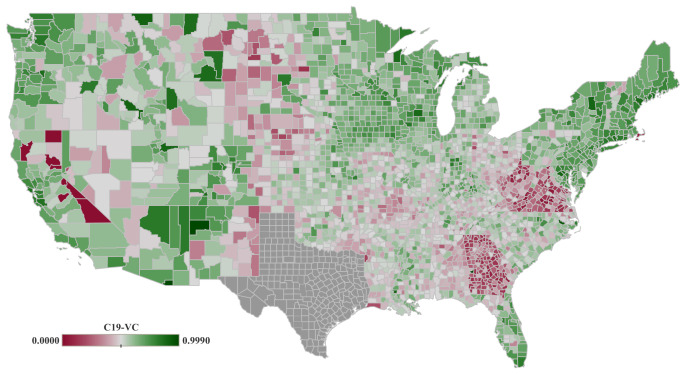
Percentage of the population completely vaccinated (C19-VC) for the period from 2 to 14 June 2020. Data is spatially aggregated by counties for the United States mainland. Data scores range from total population remains unvaccinated (0) to fully vaccinated (1). Note that vaccine participation data for the counties within state of Texas are not available, and is depicted in gray color in this map.

**Figure 4 ijerph-19-15716-f004:**
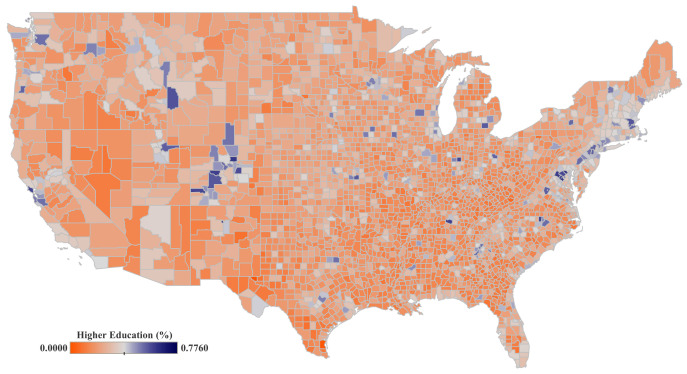
Percentage of the population with college education or above (EL-MC). Data is spatially aggregated by counties for the United States mainland. Data scores range from the minimum (0) to the maximum percentage (0.78) of people holding a high educational level.

**Figure 5 ijerph-19-15716-f005:**
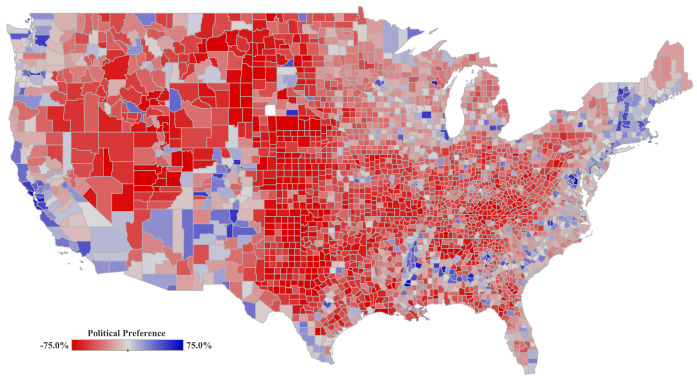
Relative difference between the winning party in the 2020 US Presidential Election. Data is spatially aggregated by counties for the United States mainland. The color legend shows the winning party and the shade indicates the relative percentage difference.

**Table 1 ijerph-19-15716-t001:** Complete list of indicators initially considered for this study. Each row corresponds to an individual indicator. The list includes 25 individual indicators that are part of 6 variable groups. We include metadata such as indicator code, variable group (cluster), and a brief description of each indicator.

CODE	Variable Group	Variable Descriptor
C19-TT	COVID-19—Epidemiological Data	COVID-19, number of tests performed
C19-CC	COVID-19—Epidemiological Data	COVID-19, number of cases
C19-DD	COVID-19—Epidemiological Data	COVID-19, number of deaths
C19-TC	COVID-19—Epidemiological Data	COVID-19, test capacity
C19-MC	COVID-19—Mask usage within 1.80 m	Combined Score
C19-MA	COVID-19—Mask usage within 1.80 m	Always
C19-MF	COVID-19—Mask usage within 1.80 m	Frequently
C19-MS	COVID-19—Mask usage within 1.80 m	Sometimes
C19-MR	COVID-19—Mask usage within 1.80 m	Rarely
C19-MN	COVID-19—Mask usage within 1.80 m	Never
C19-MO	COVID-19—Mobility	Movement change compared to baseline
C19-PT	COVID-19—Mobility	Percent people staying at home
C19-VC	COVID-19—Vaccine	Percent of people with 2 vaccine shots completed
EL-LC	Educational Level	Percent of people with less than college degree
EL-MC	Educational Level	Percent of people holding college degree or higher
EL-FI	Educational Level	Federal investment in Education
HC-HB	Health system capacity	Number of beds in hospitals
HC-HO	Health system capacity	Occupancy rate in hospitals
HC-PW	Health system capacity	Percent essential workers
EC-PO	Economic Level	Poverty estimates
EC-UN	Economic Level	Percentage of unemployment
EC-IN	Economic Level	Median of Income
PP-DW	Political Preference	Democratic Party wins
PS-PD	Population settings	Population density
PS-RL	Population settings	Percent of people living in rural areas

**Table 2 ijerph-19-15716-t002:** Complete list of indicators initially considered for this study. Each row corresponds to an individual indicator. The list includes 25 individual indicators that are part of 6 variable groups. We include additional information such as sample characteristics, time frame, and value range.

CODE	Sample (N)	Time Frame	Value Range
C19-TT	762,382 people	2–14 July 2020	Number: (0–28,282.28) tests/county
C19-CC	53,134 people	2–14 July 2020	Number: (0–2566.638) cases/county
C19-DD	613 people	2–14 July 2020	Number: (0–35.04615) deaths/county
C19-TC	574,339 people	2–14 July 2020	Number: (0–28,282.27) tests/county
C19-MC	250,000 people	2–14 July 2020	Number: (2.014–4.849) range/county
C19-MA	250,000 people	2–14 July 2020	Number: (0.115–0.889) range/county
C19-MF	250,000 people	2–14 July 2020	Number: (0.029–0.549) range/county
C19-MS	250,000 people	2–14 July 2020	Number: (0.001–0.422) range/county
C19-MR	250,000 people	2–14 July 2020	Number: (0–0.384) range/county
C19-MN	250,000 people	2–14 July 2020	Number: (0–0.432) range/county
C19-MO	33,069 people	2–14 July 2020	Number: (–0.29–0.97) units/county
C19-PT	33,069 people	2–14 July 2020	Number: (0.10–0.35) units/county
C19-VC	98,838 people	1 September 2021	Percentage: (0.3–68.36) %
EL-LC	56,039,323	2019	Percentage: (0.052–0.606) %
EL-MC	34,223,453	2019	Percentage: (0–0.776) %
EL-FI	34,223,453	2019	Number: (25,700–5,972,592,206) USD
HC-HB	24,231	2–14 July 2020	Number: (532–50,471) beds/county
HC-HO	24,231	2–14 July 2020	Rate: (0–0.915) beds/day
HC-PW	3,212,312	2–14 July 2020	Percent: (0.17–0.79) %
EC-PO	Whole US	2019	Number: (12–1,319,242) people
EC-UN	Whole US	2019	Number: (4–234,262) people
EC-IN	Whole US	2019	Number: (24,732–151,806) USD
PP-DW	Whole US	3 November 2020	Binary: (0, 1)
PS-PD	Whole US	2019	Total: (0.01–27,819.80) people/sq.mile
PS-RL	Whole US	2019	Percentage: (0–100) %

**Table 3 ijerph-19-15716-t003:** Multiple regression results for the target variable: mobility restrictions (C19-MO).

	Dependent Variable:		
	Mobility Restrictions		
	(1)	(2)	(3)
PS-PD	−1.722 ***	−1.691 ***	−1.230 ***
	(0.178)	(0.184)	(0.176)
EL-MC		−0.018	0.395 ***
		(0.027)	(0.044)
PP-DW	−0.026 ***	−0.024 ***	−0.025 ***
	(0.005)	(0.006)	(0.006)
C19-CC			−1.839 ***
			(0.141)
HC-PW			0.502 ***
			(0.057)
EC-PO	−0.236 ***	−0.253 ***	−0.230 ***
	(0.039)	(0.047)	(0.047)
HC-HO			−0.044 ***
			(0.011)
EC-UN			1.661 ***
			(0.397)
Constant	0.041 ***	0.047 ***	−0.323 ***
	(0.006)	(0.011)	(0.040)
Observations	2474	2474	2125
R${}^{2}$	0.076	0.076	0.213
Adjusted R${}^{2}$	0.074	0.074	0.210
Residual Std. Error	0.093 (df = 2470)	0.093 (df = 2469)	0.086 (df = 2116)
F Statistic	67.240 *** (df = 3; 2470)	50.531 *** (df = 4; 2469)	71.514 *** (df = 8; 2116)

Note: * *p* < 0.1; ** *p* < 0.05; *** *p* < 0.01.

**Table 4 ijerph-19-15716-t004:** Multiple regression results for the target variable: mask adoption (C19-MC).

	Dependent Variable:		
	Mask Adoption		
	(1)	(2)	(3)
PS-PD	1.793 ***	1.110 ***	0.835 ***
	(0.278)	(0.284)	(0.280)
EL-MC		0.360 ***	0.244 ***
		(0.039)	(0.061)
PP-DW	0.158 ***	0.120 ***	0.104 ***
	(0.008)	(0.009)	(0.009)
C19-CC			0.784 ***
			(0.223)
HC-PW			−0.093
			(0.075)
EC-PO	−0.199 ***	0.123 *	−0.011
	(0.055)	(0.064)	(0.070)
HC-HO			0.123 ***
			(0.017)
EC-UN			7.343 ***
			(0.583)
Constant	0.660 ***	0.545 ***	0.470 ***
	(0.008)	(0.015)	(0.051)
Observations	3072	3072	2456
R${}^{2}$	0.165	0.189	0.291
Adjusted R${}^{2}$	0.165	0.187	0.289
Residual Std. Error	0.149 (df = 3068)	0.147 (df = 3067)	0.138 (df = 2447)
F Statistic	202.784 *** (df = 3; 3068)	178.117 *** (df = 4; 3067)	125.615 *** (df = 8; 2447)

Note: * *p* < 0.1; ** *p* < 0.05; *** *p* < 0.01.

**Table 5 ijerph-19-15716-t005:** Multiple regression results for the target variable: vaccine participation (C19-VC).

	Dependent Variable:		
	Vaccine Participation		
	(1)	(2)	(3)
PS-PD	−0.584	−1.501 ***	−0.951 **
	(0.382)	(0.390)	(0.384)
EL-MC		0.484 ***	0.654 ***
		(0.053)	(0.084)
PP-DW	0.130 ***	0.079 ***	0.057 ***
	(0.011)	(0.012)	(0.012)
C19-CC			−0.904 ***
			(0.306)
HC-PW			0.231 **
			(0.103)
EC-PO	−1.102 ***	−0.669 ***	−0.832 ***
	(0.075)	(0.088)	(0.096)
HC-HO			0.054 **
			(0.023)
EC-UN			7.338 ***
			(0.799)
Constant	0.613 ***	0.458 ***	0.198 ***
	(0.011)	(0.020)	(0.071)
Observations	3072	3072	2456
R${}^{2}$	0.097	0.121	0.174
Adjusted R${}^{2}$	0.096	0.120	0.171
Residual Std. Error	0.204 (df = 3068)	0.202 (df = 3067)	0.189 (df = 2447)
F Statistic	109.844 *** (df = 3; 3068)	105.408 *** (df = 4; 3067)	64.402 *** (df = 8; 2447)

Note: * *p* < 0.1; ** *p* < 0.05; *** *p* < 0.01.

**Table 6 ijerph-19-15716-t006:** IV regression results for the target variable: mobility restrictions (C19-MO).

	Dependent Variable:		
	Mobility Restrictions		
	(1)	(2)	(3)
EL-MC	−0.708 ***	−0.834 ***	0.535
	(0.087)	(0.180)	(0.348)
PP-DW		0.028	−0.039 ***
		(0.021)	(0.013)
PS-PD			−1.050 ***
			(0.180)
C19-CC			−1.498 ***
			(0.138)
HC-PW			0.699 *
			(0.399)
EC-PO			0.054
			(0.088)
HC-HO			−0.043 ***
			(0.015)
EC-UN			0.334
			(0.536)
Constant	0.169 ***	0.193 ***	−0.473
	(0.024)	(0.042)	(0.302)
Weak instruments	0	0	0
Wu-Hausman	0	0	0.3
Observations	1140	1129	1054
Residual Std. Error	0.100 (df = 1138)	0.106 (df = 1126)	0.075 (df = 1045)

Note: * *p* < 0.1; ** *p* < 0.05; *** *p* < 0.01.

**Table 7 ijerph-19-15716-t007:** IV regression results for the target variable: mask adoption (C19-MC).

	Dependent Variable:		
	Mask Adoption		
	(1)	(2)	(3)
EL-MC	1.101 ***	0.720 ***	0.333
	(0.129)	(0.225)	(0.561)
PP-DW		0.089 ***	0.102 ***
		(0.026)	(0.021)
PS-PD			0.489
			(0.301)
C19-CC			0.418 *
			(0.227)
HC-PW			−0.087
			(0.637)
EC-PO			−0.335 **
			(0.149)
HC-HO			0.088 ***
			(0.024)
EC-UN			7.859 ***
			(0.855)
Constant	0.426 ***	0.500 ***	0.504
	(0.035)	(0.052)	(0.483)
Weak instruments	0	0	0
Wu-Hausman	0	0.12	0.5
Observations	1167	1160	1078
Residual Std. Error	0.148 (df = 1165)	0.135 (df = 1157)	0.125 (df = 1069)

Note: * *p* < 0.1; ** *p* < 0.05; *** *p* < 0.01.

**Table 8 ijerph-19-15716-t008:** IV regression results for the target variable: vaccine participation (C19-VC).

	Dependent Variable:		
	Vaccine Participation		
	(1)	(2)	(3)
EL-MC	0.706 ***	0.531	1.774 **
	(0.170)	(0.324)	(0.863)
PP-DW		0.042	0.024
		(0.038)	(0.032)
PS-PD			−0.821 *
			(0.462)
C19-CC			−0.970 ***
			(0.349)
HC-PW			1.541
			(0.979)
EC-PO			−0.518 **
			(0.230)
HC-HO			0.043
			(0.037)
EC-UN			9.018 ***
			(1.314)
Constant	0.353 ***	0.386 ***	−0.802
	(0.046)	(0.075)	(0.742)
Weak instruments	0	0	0
Wu-Hausman	0.6	0.98	0.09
Observations	1171	1160	1078
Residual Std. Error	0.195 (df = 1169)	0.195 (df = 1157)	0.192 (df = 1069)

Note: * *p* < 0.1; ** *p* < 0.05; *** *p* < 0.01.

**Table 9 ijerph-19-15716-t009:** Comparison between OLS and IV regressions for all target variables.

	Dependent Variable:					
	Mobility Restrictions		Mask Adoption		Vaccine Participation	
	OLS	Instrumental	OLS	Instrumental	OLS	Instrumental
		Variable		Variable		Variable
	(1)	(2)	(3)	(4)	(5)	(6)
PS-PD	−1.230 ***	−1.050 ***	0.835 ***	0.489	−0.951 **	−0.821 *
	(0.176)	(0.180)	(0.280)	(0.301)	(0.384)	(0.462)
EL-MC	0.395 ***	0.535	0.244 ***	0.333	0.654 ***	1.774 **
	(0.044)	(0.348)	(0.061)	(0.561)	(0.084)	(0.863)
PP-DW	−0.025 ***	−0.039 ***	0.104 ***	0.102 ***	0.057 ***	0.024
	(0.006)	(0.013)	(0.009)	(0.021)	(0.012)	(0.032)
C19-CC	−1.839 ***	−1.498 ***	0.784 ***	0.418 *	−0.904 ***	−0.970 ***
	(0.141)	(0.138)	(0.223)	(0.227)	(0.306)	(0.349)
HC-PW	0.502 ***	0.699 *	−0.093	−0.087	0.231 **	1.541
	(0.057)	(0.399)	(0.075)	(0.637)	(0.103)	(0.979)
EC-PO	−0.230 ***	0.054	−0.011	−0.335 **	−0.832 ***	−0.518 **
	(0.047)	(0.088)	(0.070)	(0.149)	(0.096)	(0.230)
HC-HO	−0.044 ***	−0.043 ***	0.123 ***	0.088 ***	0.054 **	0.043
	(0.011)	(0.015)	(0.017)	(0.024)	(0.023)	(0.037)
EC-UN	1.661 ***	0.334	7.343 ***	7.859 ***	7.338 ***	9.018 ***
	(0.397)	(0.536)	(0.583)	(0.855)	(0.799)	(1.314)
Constant	−0.323 ***	−0.473	0.470 ***	0.504	0.198 ***	−0.802
	(0.040)	(0.302)	(0.051)	(0.483)	(0.071)	(0.742)
Weak instruments		0		0		0
Wu-Hausman		0.3		0.5		0.09

Note: * *p* < 0.1; ** *p* < 0.05; *** *p* < 0.01.

## Data Availability

All the data used in this manuscript are compiled and freely available for users in the GitHub repository: https://github.com/mprtrmrtz/maskmandate (accessed on 9 October 2022).

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
