# Peer review of "Social Behavior and COVID-19: Analysis of the Social Factors behind Compliance with Interventions across the United States"

_ijerph, 2022, doi:10.3390/ijerph192315716_

Round 1

Reviewer 1 Report

This is a interesting and data-heavy analysis of geographical variation in responses to covid-19 at county level in the US during the first wave of the pandemic. It presents a large number of correlations with little attempt to discuss causal mechanisms. For example, (a) some of the variables are standing for differences in perceptions (e.g understanding of the reasons for restrictions), while others may be associated with opportunities (e.g. possibilities of distancing may differ by occupation and income level). A discussion of possible causal models could motivate the selection of explanatory variables (as they stand their chief justification is that they were easily available). Are any key influences missing and if so, what is the likely impact of their omission? (b) the use of federal expenditure on education as an IV has important effects; yet little discussion is devoted as to why and whether that would be an appropriate instrument. (c) political association is given prominence, which chimes with US practice. However this can hardly be thought an independent variable and is obviously a shorthand for other associations and missing variables.  

Nevertheless, the paper deserves to be published - with some additions in the discussion. I also think that it would benefit from drawing from and connecting more with international evidence and discussion.

Reviewer 2 Report

The paper contributes to understand the correlation between different societal variables along with healthcare capacity related variables with regards to peoples level of compliance with three main governmental mandates in the United States: mobility restrictions, mask adoption, and vaccine participation.

1. The importance and relevance of this paper concern the correlation analysis between societal variables and mobility restrictions, mask adoption, and vaccine participation using multi linear regression model an instrumental variable analysis.

2. The model presented in the paper may not be meaningful because the explanatory power of the model is low and non-significant variables are also mixed. This is considered to be caused by the discrepancy between the data and the model that appears in trying to explain people's microscopic behavior with macroscopic data.

3. It is analyzed that the influence of various independent variables on the three dependent variables is different, and it is believed that it is in setting a good response direction to COVID-19 accordingly.

4. The effects of independent variables on dependent variables are not well explained in the results of various models. It is questionable whether the results of the model do not conform to common sense or are difficult to explain properly.

5. I consider that the results of the analysis are not properly explained in the discussion. It is also necessary to present implications based on the research results.

6. I consider references are appropriate.

Reviewer 3 Report

Congratulations on the relevance of the paper. It is an excellent contribution to a better understanding of social and territorial differences in terms of how to deal with the restrictions imposed by the pandemic. On the other hand, it also helps to explain the impact of the disease on different communities.

The article includes very detailed data collection and processing, which I value in empirical studies. Furthermore it considers the spatial distribution of information. For this reason, it would be interesting to consider applying spatial econometric models, which I recommend for future work. Feel free to contact me if you need some help about this.

Some observations:

- in line 283, where is Table A1, must be Figure A1

Round 2

Reviewer 2 Report

The paper was well revised.